# Dynamic anti-correlations of water hydrogen bonds

Lucas Gunkel [1,3], Amelie A. Ehrhard[1,3], Carola S. Krevert [1], Bogdan A. Marekha [1,2], Mischa Bonn [1], Maksim Grechko[1] & Johannes Hunger [1] ✉

Water is characterized by strong intermolecular hydrogen bonds (H-bonds) between molecules. The two hydrogen atoms in one water molecule can form H-bonds of dissimilar length. Although intimately connected to water's anomalous properties, the details and the origins of the asymmetry have remained elusive. We study water's H-bonds using the O-D stretching vibrations as sensitive reporters of H-bonding of $D_2O$ and HOD in dimethylformamide. Broader inhomogeneous linewidths of the OD band of HOD compared to the symmetric and asymmetric OD stretching modes of $D_2O$ together with density functional theory calculations provide evidence for markedly anti-correlated H-bonds: water preferentially forms one weak and one strong H-bond. Coupling peaks in the spectra for $D_2O$ directly demonstrate anti-correlated H-bonds and these anti-correlations are modulated by thermal motions of water on a sub-picosecond timescale. Experimentally inferred H-bond distributions suggest that the anti-correlations are a direct consequence of the H-bonding potential of $XH_2$ groups, which we confirm for the $ND_2$ group of urea. These structural and dynamic insights into H-bonding are essential for understanding the relationship between the H-bonded structure and phase behavior of water.

The peculiar properties of water have been ascribed to its intermolecular interactions: strong and directional hydrogen bonds (H-bonds), which determine its three-dimensional structure. Albeit controversially discussed[1,2], the traditional picture of liquid water forming a symmetric coordination structure (Fig. 1a)[3] has been challenged by X-ray spectroscopy[4,5]. Indeed, molecular dynamics simulations have shown that symmetric/tetrahedral coordination cannot capture all structural details of liquid water[6–9]. Transient deviations from on-average symmetric coordination are intimately connected to local correlations of the H-bond strengths[7,8,10] and asymmetric coordination geometries with, e.g., one water molecule forming two strong and two weak H-bonds result in ring- or chain-like structures of water[1,6,9]. Such motifs have been suggested to have profound implications for the phase behavior of water[10] and may explain some of the anomalous

properties of water at reduced temperatures, such as a density maximum at 277 K or a nonlinear temperature-dependence of the heat capacity[11,12]. Also, at ambient temperatures, where the H-bonded structure determines water's performance as solvent[13], asymmetric coordination geometries exist[14,15], yet, thermal fluctuations seem to limit the spatial extent of such structural correlations[16]. However, experimental evidence on the exact details of the H-bond symmetry, such as the origin of the asymmetry, correlation of H-bond lengths, and their evolution, is lacking.

The frequencies and linewidths of O-H stretching vibrations reflect the length and symmetry of water's H-bonds[17]. The frequency $\tilde{\nu}_l$ of a single (local) O-H stretching oscillator depends on H-bonding distance $d_{H\text{-}O}$ (Fig. 1b)[18]. A water molecule has two local O-H stretching modes ($l1$, $l2$), which interact to form symmetric (*sym*) and asymmetric

[1]Max-Planck Institute for Polymer Research, Ackermannweg 10, Mainz, Germany. [2]Present address: ENS de Lyon, CNRS, LCH, UMR 5182, 69342, Lyon cedex 07, France. [3]These authors contributed equally: Lucas Gunkel, Amelie A. Ehrhard. ✉e-mail: hunger@mpip-mainz.mpg.de

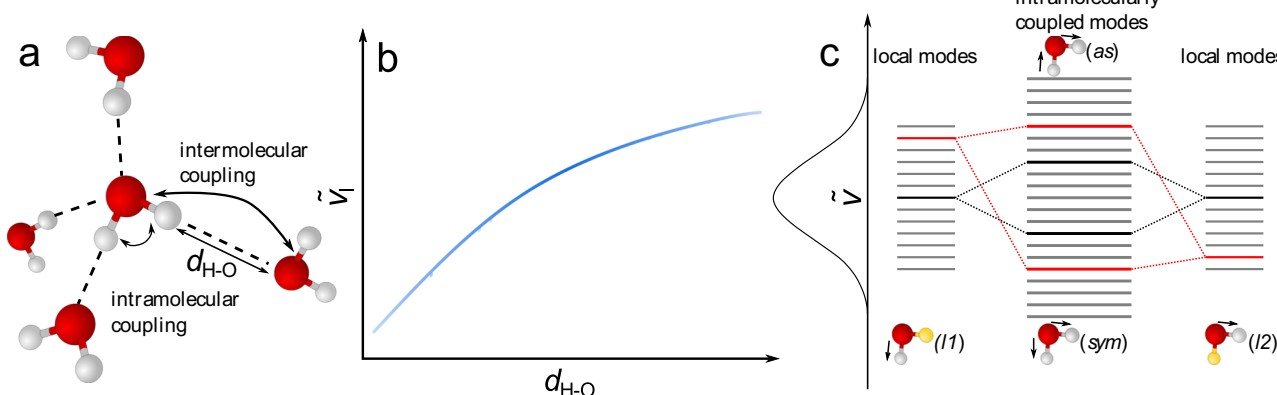

**Fig. 1 | H-bonding and vibrational structure of water. a** Schematic illustration of a (distorted) tetrahedral coordination geometry of water, with arrows indicating intra- and intermolecular coupling of O-H oscillators. The H-bond distance $d_{H-O}$ markedly affects the resonance frequency of a single O-H oscillator, as schematically depicted in (**b**), giving rise to a broad distribution of O-H stretching frequencies in water. The experimentally observed O-H stretching linewidth (depicted as vertical Gauss distribution in **c**) is further affected by vibrational coupling, as illustrated for intramolecular coupling in (**c**).

(*as*) vibrations. The instantaneous frequencies of *sym* and *as* depend on the instantaneous frequencies of *l1* and *l2*, thus on the H-bond distances of the two O-H groups, and on the coupling strength (Fig. 1c). These different factors governing the instantaneous frequencies of *sym*, *as*, *l1* and *l2* make also the linewidths of coupled and uncoupled modes to differ. In turn, the correlation of H-bonds can be probed via the lineshapes of *sym* and *as*. However, O-H stretching frequencies in neat H₂O—and similarly O-D stretching frequencies in D₂O—are also affected by intermolecular coupling with O-H groups of surrounding molecules (Fig. 1a)[19,20]. This coupling gives rise to vibrational excitons—O-H stretching modes delocalized over several molecules[19-22]. This delocalized character of the O-H stretching band in neat water impedes the direct correlation of spectral and structural information. To eliminate delocalization, we dilute water in dimethylformamide (DMF), isolating water molecules from each other[23-27].

The broadening of the distribution of O-H stretching frequencies due to varying H-bond length is represented by inhomogeneous broadening. To reveal this broadening, we use two-dimensional infrared (2D-IR) spectroscopy[28]. In conjunction with density functional theory (DFT) calculations, we show that the linewidths of the coupled and local O-D stretching modes of isolated water in DMF indeed contain the correlation of H-bond lengths. Coupling peaks in the spectra reveal that the two H-bonds of a D₂O molecule are anti-correlated, but this anti-correlation persists for only a few hundred femtoseconds. Comparison of experimentally inferred H-bond distributions to the DFT-calculated H-bonding potential indicates that anti-correlations are intrinsic to the H-bonding potential of XD₂ groups. We verify this hypothesis with experiments on the ND₂ groups of urea. As such, our results evidence a strong, yet rapidly randomized, H-bond anti-correlation for molecular entities that can donate two H-bonds. Our observations suggest that these anti-correlations also exist in neat water, which implies a dynamic picture with short-lived asymmetric water molecules rather than persistent chain- or ring-like structures.

## Results and discussion

To elucidate the H-bonding of water, we measure the O-D stretching vibration of 5 mol% of water in DMF, where water-water H-bonds are negligible[23]. We characterize the distribution of O-D stretching frequencies of isolated O-D groups in DMF solution using isotopically diluted water containing ~1% D₂O, ~81% H₂O, and ~18% HOD molecules. The corresponding local O-D stretching mode (*l*) of predominantly HOD molecules, is centered at $\tilde{\nu}_l$~ 2580 cm⁻¹ with a linewidth (full width at half maximum, FWHM) of ~94 cm⁻¹ (Fig. 2a, see also Supplementary

Note 1, Supplementary Fig. 1, and Supplementary Table 1). Conversely, for D₂O in DMF, intramolecular coupling of O-D oscillators produces the symmetric stretching band at $\tilde{\nu}_{sym}$~ 2540 cm⁻¹ and the asymmetric stretching band at $\tilde{\nu}_{as}$~ 2640 cm⁻¹ (Fig. 2a). Notably, *as* and *sym* have an appreciably narrower linewidth (FWHM of ~78 cm⁻¹) than the local mode *l*.

To disentangle the homogeneous and inhomogeneous broadening contributions[29] to the linear spectra (Fig. 2a), we perform 2D-IR spectroscopy experiments. Conceptually, in a 2D-IR experiment, a subset of vibrational modes resonant with a pump frequency ($\tilde{\nu}_{pump}$) is excited and the response of these oscillators is probed over a broad detection frequency range ($\tilde{\nu}_{probe}$) by a probe pulse[28]. The response typically contains positive signals from induced absorption due to the excited state absorption, and negative signals due to ground state bleaching and stimulated emission at the fundamental transition. Variation of $\tilde{\nu}_{pump}$ results in a two-dimensional spectrum reflecting the correlation between excited and detected vibrational frequencies. For inhomogeneously broadened bands the bleaching signal is elongated along the diagonal, with an adjacent, red-shifted induced absorption, as shown for HOD in DMF in Fig. 2b. The widths of these signals perpendicular to the diagonal are determined solely by homogeneous broadening[28] and the diagonal width is given by homogeneous and inhomogeneous broadening. For D₂O in DMF, two pairs of bleaching/induced absorption signals are present for *sym* and *as* at the diagonal (Fig. 2c). Additionally, we observe an off-diagonal bleaching signal at ($\tilde{\nu}_{pump} \approx 2540$ cm⁻¹/ $\tilde{\nu}_{probe} \approx 2640$ cm⁻¹), indicating coupling between *sym* and *as*. The coupling signal above the diagonal ($\tilde{\nu}_{pump} \approx 2640$ cm⁻¹/ $\tilde{\nu}_{probe} \approx 2540$ cm⁻¹) presumably overlaps with the induced absorption of *as* and, thus, is not visible in the spectrum. These off-diagonal signals will be discussed in more detail below.

For a quantitative discussion of the linewidths, we infer the homogenous linewidth by fitting a sum of two Lorentzians of opposite signs to the anti-diagonal cuts (Fig. 2d, e), yielding homogeneous linewidths of 63 cm⁻¹ (*l*), 41 cm⁻¹ (*sym*), and 47 cm⁻¹ (*as*) (see Supplementary Note 2 and Supplementary Table 2). These homogeneous linewidths are likely overestimated due to ultrafast spectral diffusion dynamics. Nevertheless, they show that the differing line widths in Fig. 2a stem partly from differing homogeneous broadening. With these homogeneous widths, we deconvolve the inhomogeneous and homogeneous contributions to the diagonal widths: We constrain the homogeneous width to the values from the anti-diagonals and model the diagonal cuts with a Voigt profile (a Gaussian inhomogeneous distribution convolved with the Lorentzian homogeneous band[28,30,31]) to derive the inhomogeneous broadening (Fig. 2d, e, for details see Supplementary Note 2

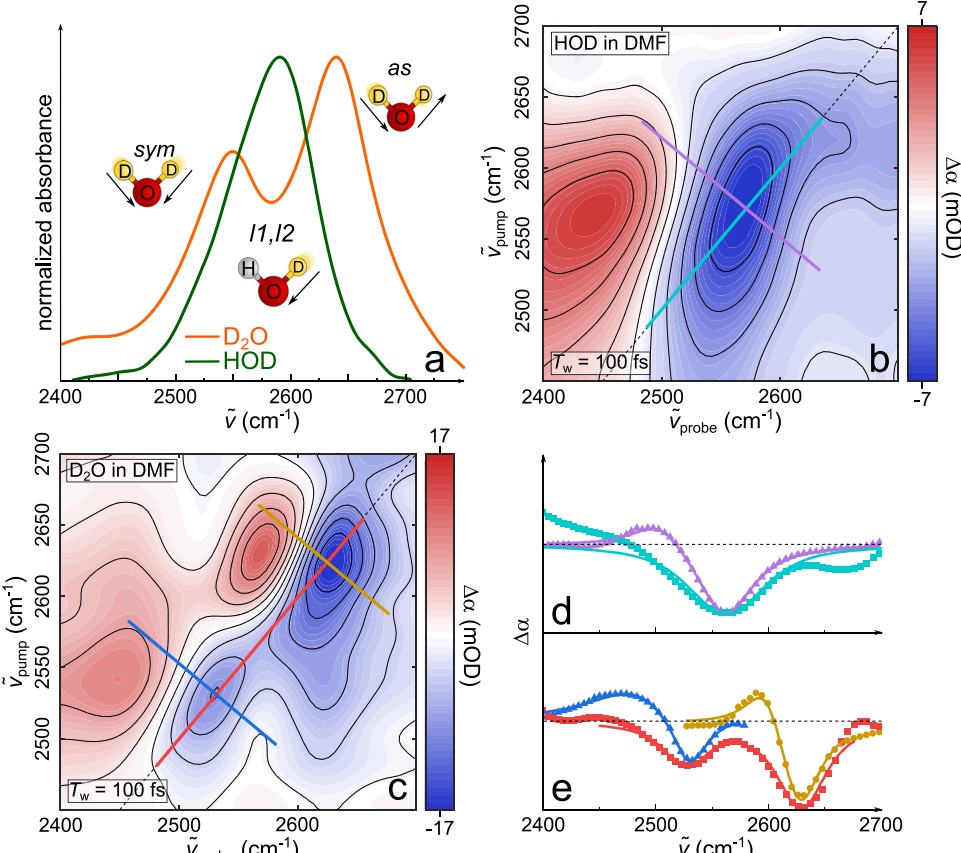

**Fig. 2 | Isolating inhomogeneous contributions to the O-D stretching line-widths. a** Solvent-background subtracted, normalized IR absorption spectra of 5 mol% water in DMF: $D_2O$ (orange) and HOD (green). Isotropic 2D-IR spectra at waiting time $T_w = 100$ fs of (**b**) HOD in DMF (10% $D_2O$ in $H_2O$) and (**c**) $D_2O$ in DMF. Note that distortions of the 2D-IR lineshape in (**b**) at $\tilde{\nu}_{pump} \approx 2650$ cm$^{-1}$ presumably result from residual $D_2O$. Diagonal and anti-diagonal profiles along the lines indicated in the 2D spectra (**b**, **c**, colored lines) are displayed for HOD in (**d**) and for $D_2O$ in (**e**). Symbols in (**d**, **e**) show experimental data and solid lines show fits (for details, see Supplementary Note 2). Dotted line represents $\Delta\alpha = 0$. Note that the linewidths of the diagonal signals in (**d**, **e**) differ from the widths in (**a**) due to the different experimental sensitivities to the transition dipole[28].

and Supplementary Tables 2–5). The thus obtained purely inhomogeneous width $\Gamma_{G,i}$ of the local O-D stretching mode of HOD ($\Gamma_{G,l} = 38$ cm$^{-1}$) is about twofold broader than the inhomogeneous widths of *sym* and *as* ($\Gamma_{G,sym} = 20$ cm$^{-1}$, $\Gamma_{G,as} = 18$ cm$^{-1}$). As such, although $D_2O$ and HOD experience very similar H-bonding environments, coupling of the two O-D stretching modes in $D_2O$ results in a narrower distribution of frequencies for *as* and *sym* as compared to *l*, which is the predominant cause for the differing widths in Fig. 2a.

To understand the origin of these different inhomogeneous widths, we calculated normal mode frequencies under the harmonic approximation for different H-bond geometries using DFT. We specifically address the effect of coupling on the harmonic normal mode frequencies by calculating the frequencies for *l*, *sym*, and *as* using H/D isotope exchange: We optimized the geometry of a water molecule donating two H-bonds to two DMF molecules, with the H-bond lengths (H/D$_{water}$ – O$_{DMF}$ distance) constrained to typical H-bonding distances[23] ranging from 1.5 to 2.4 Å. The resulting frequency maps for D$^{(1)}$-O-D$^{(2)}$, D$^{(1)}$-O-H$^{(2)}$, and H$^{(1)}$-O-D$^{(2)}$ molecules as a function of H/D$_{water}$ – O$_{DMF}$ distances $d_1$ and $d_2$ for both H/D atoms of water, where super- and subscripts denote the two light atoms of water, are displayed in Fig. 3a. This allows us to assess the effect of (thermal) fluctuations of H-bond lengths on the vibrational frequencies of *sym*, $\tilde{\nu}_{sym}$, the local modes of O-D$^{(1)}$/O-D$^{(2)}$, $\tilde{\nu}_{l1}$/ $\tilde{\nu}_{l2}$, and of *as*, $\tilde{\nu}_{as}$. In particular, we consider three limiting cases of H-bond fluctuations: (i) directly correlated ($d_1 = d_2$, Fig. 3b), (ii) anti-correlated ($d_1 + d_2 = 3.8$ Å, Fig. 3c), and (iii) uncorrelated ($d_2 = 1.9$ Å, Fig. 3d) H-bond distances.

These limiting cases show that $\tilde{\nu}_{l1}$ and $\tilde{\nu}_{l2}$ simply scale with H-bond distances $d_1$ and $d_2$, respectively (see also Fig. 1b). When $\tilde{\nu}_{l1}$ and $\tilde{\nu}_{l2}$ are dissimilar, the frequencies of the 'coupled' modes $\tilde{\nu}_{sym}$ and $\tilde{\nu}_{as}$ are close to those of the local modes $\tilde{\nu}_{l1}$ or $\tilde{\nu}_{l2}$ (see e.g., $d_1 < 1.8$ Å or $d_1 > 2.0$ Å in Fig. 3c, d). For H-bond configurations for which $d_1 \approx d_2$, coupling of the local modes $\tilde{\nu}_{l1}$ and $\tilde{\nu}_{l2}$ results in a separation of frequencies for the coupled modes $\tilde{\nu}_{sym}$ and $\tilde{\nu}_{as}$ (Figs. 3b and 1.8 Å $< d_1 < 2.0$ Å in Fig. 3c, d). Therefore, all frequencies show a similar dependence on $d_1$ for case (i) of directly correlated H-bonds. For cases (ii) anti-correlated and (iii) uncorrelated H-bonds, coupling gives rise to avoided crossings[32], and makes $\tilde{\nu}_{sym}$ and $\tilde{\nu}_{as}$ to not just scale with $d_1$. Consequently, for a given range of thermally accessible H-bond distances $d_1$, coupling results in a narrower distribution of $\tilde{\nu}_{sym}$ and $\tilde{\nu}_{as}$ as compared to $\tilde{\nu}_{l1}$ and $\tilde{\nu}_{l2}$ for (ii) and (iii). Conversely, for (i), the range of thermally accessible frequencies is similar for all vibrations. As the spread of H-bond geometries underlies the inhomogeneous line-width of these modes, (ii) anti-correlated and (iii) uncorrelated H-bond geometries can qualitatively explain the experimentally observed reduced $\Gamma_{G,sym}$ and $\Gamma_{G,as}$, relative to $\Gamma_{G,l}$. For instance, assuming thermally accessible H-bond distances ranging from 1.8 Å to 2.0 Å, the data in Fig. 3 suggest the resulting inhomogeneous linewidths of *sym* and *as* to be ~2 and ~4 times, respectively, narrower than that of *l* (HOD). Hence, while the comparison between calculated and experimental linewidths cannot discriminate between (ii) and (iii), we exclude directly correlated fluctuations of H-bond distances.

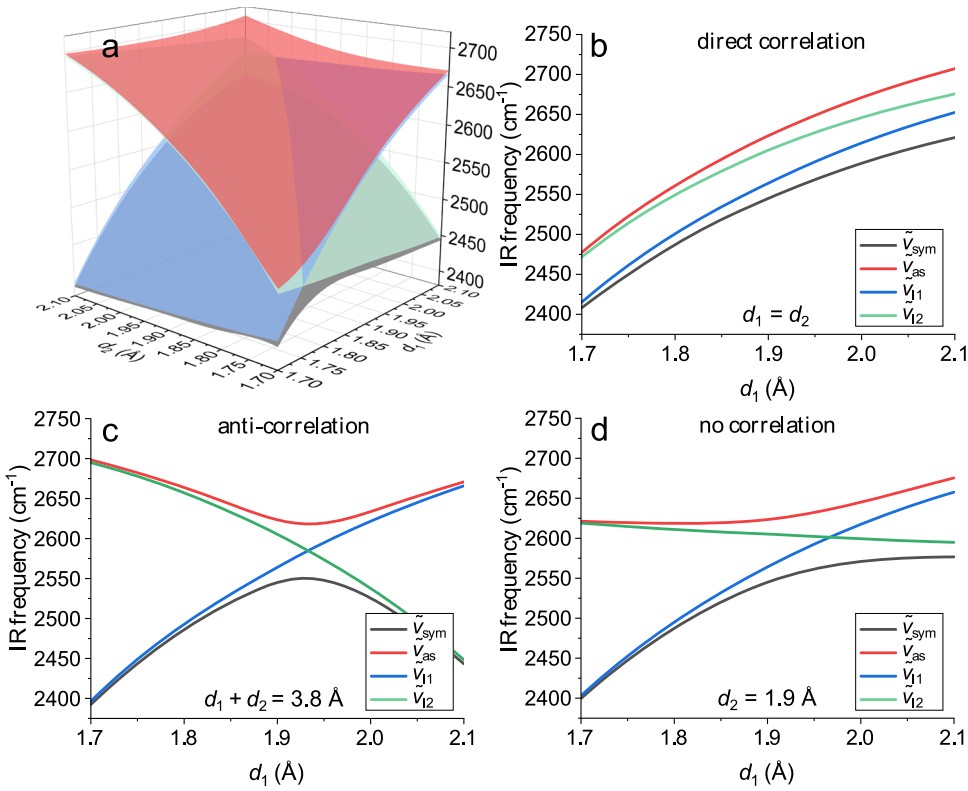

**Fig. 3 | DFT-calculated harmonic frequencies to reveal the effect of H-bond distance correlations on vibrational frequencies. a** Vibrational frequency maps for the symmetric $\tilde{\nu}_{sym}$ (gray), local $\tilde{\nu}_{l1}$ (blue), $\tilde{\nu}_{l2}$ (green), and asymmetric $\tilde{\nu}_{as}$ (red) O-D stretching modes as a function of H-bond distances $d_1$ and $d_2$, as obtained from the harmonic frequencies of relaxed surface scans using DFT calculations. **b** Diagonal cut at $d_1 = d_2$ through the frequency maps representing directly correlated H-bonds, (**c**) anti-diagonal cut at $d_1 + d_2 = 3.8$ Å representing anti-correlated H-bonds, and (**d**) cut at $d_2 = 1.9$ Å representing uncorrelated H-bond distances.

Although it is challenging to discriminate between (ii) and (iii) solely based on the frequencies of the three modes as a function of H-bond lengths, both limiting cases can be discerned by considering the relation between the DFT-calculated frequencies $\tilde{\nu}_{sym}$ and $\tilde{\nu}_{as}$: For (ii) the slopes of $\tilde{\nu}_{sym}(d_1)$ and $\tilde{\nu}_{as}(d_1)$ have opposite sign (Fig. 3c): upon increasing $d_1$, $\tilde{\nu}_{sym}$ increases (decreases) and $\tilde{\nu}_{as}$ decreases (increases) for $d_1 \leq 1.9$ Å ($d_1 \geq 1.9$ Å). Conversely, for (iii) $\tilde{\nu}_{sym}(d_1)$ and $\tilde{\nu}_{as}(d_1)$ have positive or zero slopes throughout (Fig. 3d). Hence, for anti-correlated distances $d_1$ and $d_2$ (ii), the frequencies $\tilde{\nu}_{sym}$ and $\tilde{\nu}_{as}$ are anti-correlated, while for uncorrelated distances (iii) $\tilde{\nu}_{sym}$ and $\tilde{\nu}_{as}$ are correlated. The correlation between $\tilde{\nu}_{sym}$ and $\tilde{\nu}_{as}$ can be directly interrogated with 2D-IR spectroscopy as coupling between *sym* and *as* gives rise to the cross-peak ($\tilde{\nu}_{pump} \approx 2540$ cm⁻¹/ $\tilde{\nu}_{probe} \approx 2620$ cm⁻¹) in Fig. 2c and the line shape of the cross-peak provides information on the correlations of H-bonding distances, analogously to earlier studies by Hochstrasser and coworkers[33–36] on different molecular systems. Due to the relative orientation of the transition dipoles of *sym* and *as* (~90–105°), such cross-peaks are more intense (relative to the diagonal signals) for perpendicular excitation and probing polarizations (Fig. 4a, b)[28]. The presence of the cross-peak at early times ($T_w$, delay between pump and probe pulses) and evolution of its intensity similar to the diagonal signals evidences that the cross-peak is due to coupling (Supplementary Note 3, Supplementary Table 6, and Supplementary Figs. 2–3)[28]. To identify frequency-frequency (anti-)correlations, we calculate the signal-weighted local Pearson correlation coefficient (Supplementary Note 4). At $T_w = 100$ fs (for other waiting times, see Supplementary Fig. 4) these coefficients are positive for the diagonal signals, as expected for inhomogeneously broadened bands. Conversely, the off-diagonal peak at $\tilde{\nu}_{pump} \approx 2540$ cm⁻¹/ $\tilde{\nu}_{probe} \approx 2620$ cm⁻¹ is dominated by negative correlation coefficients, suggesting anti-correlated $\tilde{\nu}_{sym}$ and $\tilde{\nu}_{as}$ (Fig. 4a). Similarly, the center line slope (CLS)[37] of −0.05 for the

cross-peak at $T_w = 100$ fs demonstrates anti-correlations between $\tilde{\nu}_{sym}$ and $\tilde{\nu}_{as}$[37]. The data in Fig. 3 show that this anti-correlation of $\tilde{\nu}_{sym}$ and $\tilde{\nu}_{as}$ signifies anti-correlated H-bond distances $d_1$ and $d_2$. Therefore, the cross-peak provides direct evidence for anti-correlated H-bond distances.

The frequency-frequency anti-correlations are however rather short-lived and are at e.g., $T_w = 250$ fs much less pronounced. The CLS at $T_w = 250$ fs is even slightly positive (Fig. 4b, for other waiting times see Supplementary Fig. 4). Quantitatively, the waiting-time dependent CLS($T_w$) of the cross-peak (Fig. 4c) rapidly decays to ~0, indicating that thermal fluctuations rapidly randomize the anti-correlation between $\tilde{\nu}_{sym}$ and $\tilde{\nu}_{as}$. Remarkably, CLS($T_w$) appears to decay with marked oscillatory dynamics. It has previously been reported that such oscillatory dynamics may stem from coherence transfer within the time interval between the two excitation pulses in the time-domain 2D-IR experiment[38,39]. This scenario is, however, rendered unlikely for the present system because (i) we find no evidence for oscillations in the 2D-IR signal intensities (Supplementary Fig. 3), (ii) 2D-IR spectra with suppressed coherence transfer from *as* to *sym*[40] exhibit the same oscillatory CLS dynamics (Supplementary Fig. 5), and (iii) also the CLS dynamics of the diagonal peak for *l* exhibit oscillatory dynamics with the same oscillation period (Supplementary Fig. 6). As such, these observations suggest that the oscillations of the frequency-frequency correlations in Fig. 4c rather originate from a modulation of the CLS due to the inherent dynamics of water, which is also supported by molecular dynamics simulations[27]. In fact, the data in Fig. 4c are well-described as a damped oscillation with an oscillation period of ~310 fs and a ~50 fs exponential decay (Fig. 4c, Supplementary Note 5, and Supplementary Table 7), qualitatively similar to predictions by simulations for water in acetonitrile[27]. These timescales are close to the characteristic timescales of the H-bond stretching vibration for neat

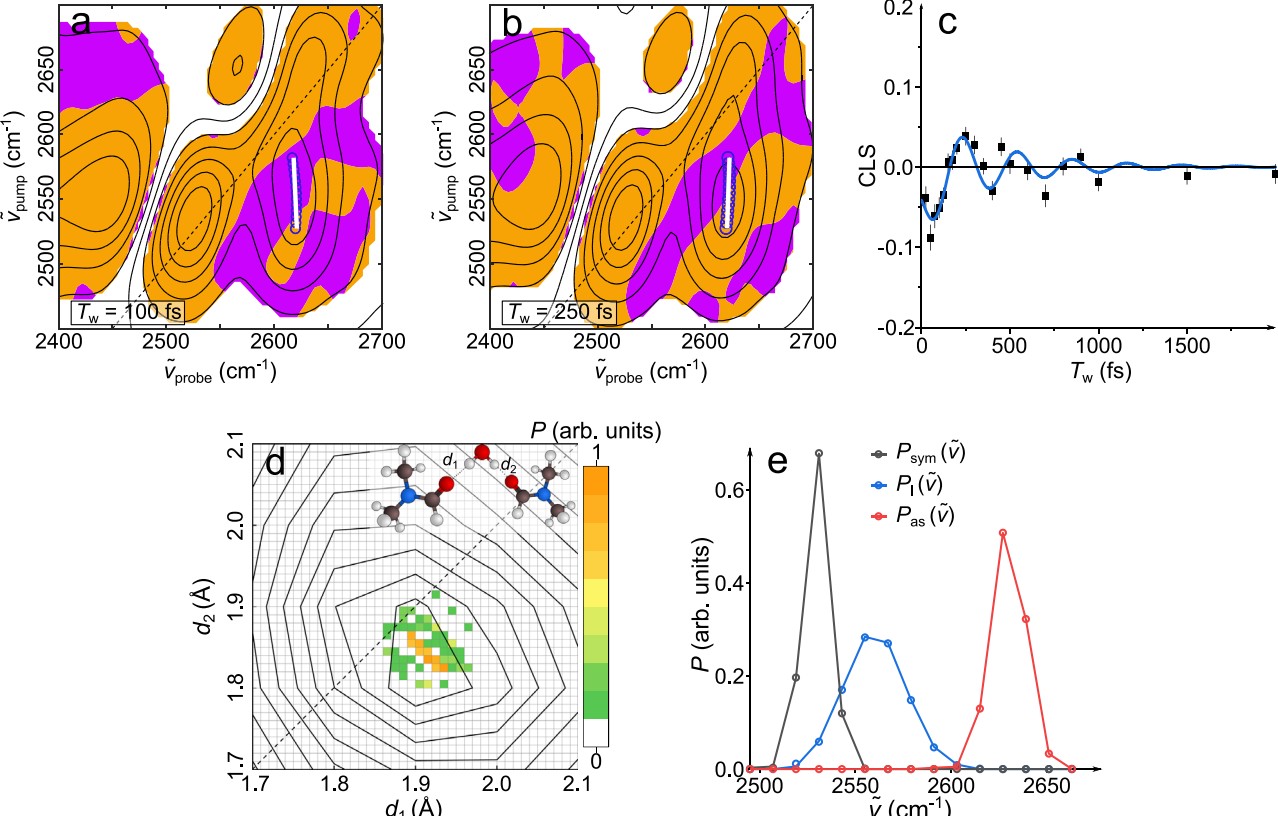

**Fig. 4 | Coupling peaks reveal dynamics of correlations and inhomogeneous linewidths demonstrate anti-correlated distributions.** Perpendicular (<XXZZ>) 2D-IR spectrum of $D_2O$ in DMF at (**a**) $T_w = 100$ fs and (**b**) $T_w = 250$ fs. Contour lines show signal intensities, orange and purple areas indicate spectral regions with positive and negative local Pearson correlation coefficients, respectively. Open blue symbols indicate the center-line position of the off-diagonal peak and the solid white line the center line. **c** Center-line slope dynamics, $CLS(T_w)$ of the off-diagonal peak. Symbols show experimental data and error bars show the uncertainty of the slope obtained from linear regression of the center points. The solid line shows a fit using a sum of a damped oscillation (~310 fs oscillation period, 470 fs damping

time) and an exponential decay (50 fs decay time). The error bars are the standard errors of the linear regression of the center lines. **d** Fitted H-bond conformation distribution $P(d_1, d_2)$ of $D_2O$ in DMF (pixel map) obtained from minimizing deviations between the experimental, discretized inhomogeneous distribution of frequencies $P_i(\tilde{\nu})$ (symbols in **e**) and the calculated distribution using $P(d_1, d_2)$ together with the frequency maps in Fig. 3a (solid lines in **e**). Contour lines in (**d**) show lines of equal population probability following the Boltzmann distribution (at 295 K) of H-bond conformations of $D_2O + 2$ DMF as obtained from the DFT-calculated total energy.

water at ~200 cm$^{-1}$ and water's libration band at ~650 cm$^{-1}$ (~530 cm$^{-1}$ for $D_2O$)[16,41–43], which are expected to be similar for $D_2O$ in DMF (see Supplementary Note 6). These intrinsic dynamics of water also modulate the decay of the CLS of the diagonal peaks (see Supplementary Note 7 and Supplementary Fig. 6). As these lower-frequency H-bond stretching vibrations and librations are thermally excited at ambient conditions, the observed CLS dynamics show that the thermally excited low-frequency modes modulate H-bond anti-correlations.

To elucidate the origin of the H-bond anti-correlation, we estimate the distribution of H-bond conformations, $P(d_1, d_2)$, (Fig. 4d) by simultaneously using the information from all vibrational modes $l$, *sym*, *as*. Therefore, we take the purely inhomogeneous distribution of frequencies, $P_i(\tilde{\nu})$, using the center frequencies and the purely inhomogeneous linewidths ($\Gamma_{G,i}$) obtained from analysis of the diagonal and anti-diagonal lineshapes in Fig. 2 (see Supplementary Note 2 and Supplementary Tables 2–3) together with the frequencies of all modes as a function of H-bond geometry $\tilde{\nu}_i(d_1, d_2)$ (frequency maps in Fig. 3a). For numerical treatment, we discretize $P_i(\tilde{\nu})$ at intervals of 15 cm$^{-1}$ (Fig. 4e, symbols) and the population of H-bond geometries $P(d_1, d_2)$ at intervals of 0.01 Å (Fig. 4d). Starting from random distributions of H-bond conformations $P(d_1, d_2)$, we optimize $P(d_1, d_2)$ such that the distributions $P_i(\tilde{\nu})$ calculated from the conformation distribution and the maps in Fig. 3a match the discretized, purely inhomogeneous distribution of frequencies, $P_i(\tilde{\nu})$ in Fig. 4e (details on the numerical

accuracy are given in Supplementary Note 8 and Supplementary Fig. 7). The distribution of H-bond distances thus obtained are shown in Fig. 4d. We note that the slight displacement of the maximum of the distribution in Fig. 4d from the diagonal (symmetric H-bond distances) results from the simplified representation of water in DMF by only one water molecule and two DMF molecules in the DFT calculations used to obtain the frequency maps in Fig. 3a (dispersive interactions between two DMF molecules lead to symmetry breaking, see Supplementary Fig. 8). Nevertheless, the distribution of H-bond conformations in Fig. 4d confirms anti-correlated H-bond distances $d_1$ and $d_2$: $P(d_1, d_2)$ is elongated along the anti-diagonal, similar to results from molecular dynamics simulations of water[44].

To pinpoint the origin of the anti-correlated H-bond distances, we compare $P(d_1, d_2)$ (pixel plot, Fig. 4d) to the distribution expected from the total energy of the DFT calculations (contour plot, Fig. 4d). Assuming a Boltzmann distribution, anti-correlated H-bonds are also predicted solely based on the DFT-calculated energy of $D_2O + 2$ DMF: the shape of the Boltzmann distribution obtained from DFT (contour lines in Fig. 4d) and the estimated distribution $P(d_1, d_2)$ (pixel plot in Fig. 4d) agree well. The differing widths of both distributions are likely due to neglecting repulsive interactions with other molecules and overestimation of the homogeneous linewidth. Irrespective of these different widths, both distributions show that the energetically most favorable H-bond distance for one H-bond

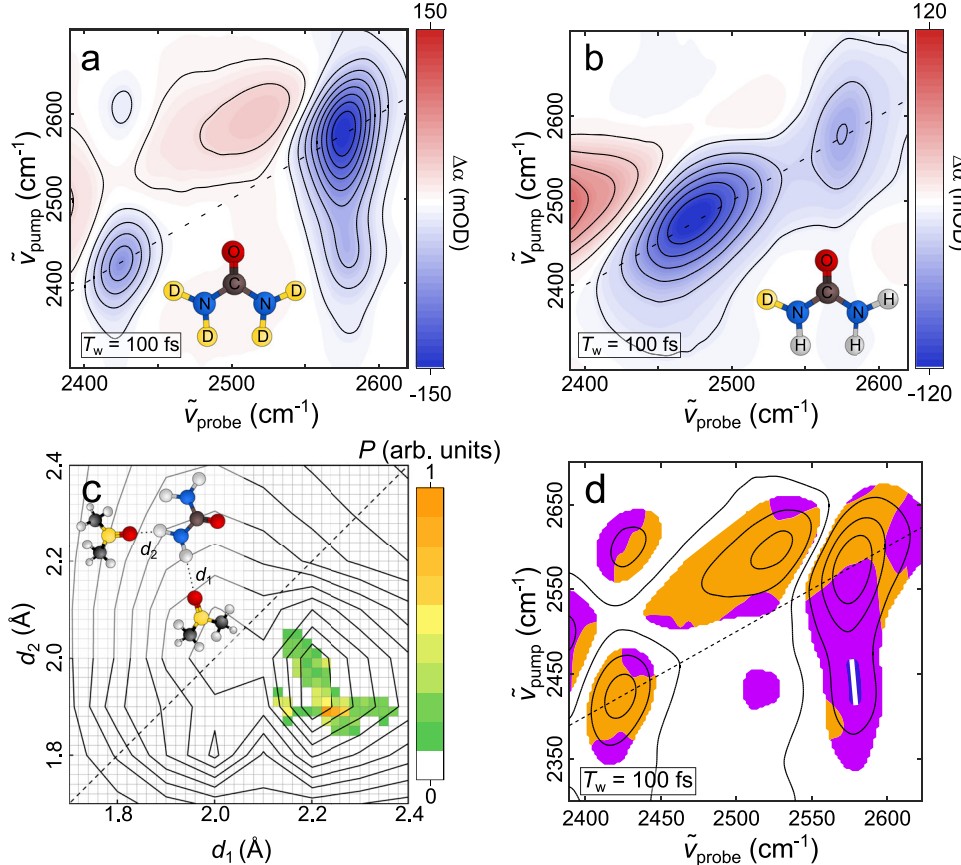

**Fig. 5 | Spectroscopic signatures of urea's N-D stretching modes reveal H-bond anti-correlations. a** Isotropic 2D-IR spectrum of urea-d4 in DMSO at $T_w = 100$ fs. **b** Isotropic 2D-IR spectrum of a 30%D/70%H isotopic mixture of urea, corresponding to ~1% urea-d4, ~8% urea-d3, ~26% urea-d2, ~41% urea-d1, and ~24% urea, in DMSO at $T_w = 100$ fs. **c** H-bond conformation distribution $P(d_1, d_2)$ of urea in DMSO (pixel map) obtained analogously to Fig. 4d together with contours representing

equal probability of the DFT energy-based Boltzmann distribution. **d** Perpendicular (<XXZZ>) 2D-IR spectrum of urea-d4 in DMSO. Contour lines show signal intensities, orange and purple areas indicate spectral regions with positive and negative local Pearson correlation coefficients, respectively. Open blue symbols indicate the center-line position of the off-diagonal peak and the solid white line shows the center line.

markedly depends on the H-bond distance of the other H-bond—the H-bonds are anti-correlated.

In fact, one might expect such distributions for any molecular group that can donate two H-bonds ($XH_2$): the H-bonding potential for such $XH_2$ fragments typically exhibits a global minimum at both H-bond equilibrium distances, with adjacent minimum energy paths for dissociation of one H-bond (see Supplementary Fig. 8). The resulting Boltzmann distributions of such potentials result in an anti-correlated distribution of H-bond distances $d_1$ and $d_2$. Thus, we hypothesize that the anti-correlated H-bond distances for $D_2O$ in DMF are intrinsic to the H-bonding potential of $XH_2$ groups. Conversely, anti-correlated H-bonds are not unique to $D_2O$ (+2 DMF) but common to $XH_2$ groups.

To test this hypothesis, we performed analogous experiments for the $N(H/D)_2$ groups of urea in dimethylsulfoxide (DMSO), for which the differences in linewidths of $l$ and $as$, $sym$ in the absorption spectra are even more pronounced (see Supplementary Fig. 9) as compared to $D_2O$ in DMF. 2D-IR spectra demonstrate that the symmetric (2420 cm$^{-1}$) and asymmetric (2560 cm$^{-1}$) N-D stretching modes (Fig. 5a) are weakly inhomogeneously broadened. Conversely, for a 30% (D) 70% (H) isotopic mixture $l$ at 2480 cm$^{-1}$ is clearly inhomogeneously broadened (Fig. 5b). Analysis of the 2D-IR lineshapes suggests the inhomogeneous linewidth of $l$ with $\Gamma_{G,l} = 47$ cm$^{-1}$ is ~50% broader than of $as$ with $\Gamma_{G,as} = 31$ cm$^{-1}$ (Supplementary Tables 4, 5). The extremely narrow linewidth of $sym$ ($\Gamma_{G,sym} = 14$ cm$^{-1}$) is presumably due to spectral distortion by a Fermi-resonance (see also Supplementary

Note 9 and Supplementary Figs. 9, 10). Nevertheless, the H-bond distributions, fitted analogously to $D_2O$ from solely $\Gamma_{G,l}$ and $\Gamma_{G,as}$ (i.e., omitting $\Gamma_{G,sym}$) and the corresponding frequency maps for urea + 4 DMSO (Supplementary Fig. 11), also exhibit anti-correlated H-bond distances (Fig. 5c). The center of this distribution is displaced from a symmetric H-bond geometry ($d_1 = d_2$), likely due to the inequivalence of the two deuterium atoms of the $ND_2$ group. Similar to our findings for $D_2O$, the elliptical distribution of H-bond geometries can again be traced to the H-bonding potential of the $ND_2$ group (Fig. 5c). Further, local correlation maps of the cross-peak at $\tilde{\nu}_{pump} \approx 2420$ cm$^{-1}$/ $\tilde{\nu}_{probe} \approx 2580$ cm$^{-1}$ in Fig. 5d evidence anti-correlated $\tilde{\nu}_{sym}$ and $\tilde{\nu}_{as}$. For urea-d4 also the cross-peak at $\tilde{\nu}_{pump} \approx 2580$ cm$^{-1}$ / $\tilde{\nu}_{probe} \approx 2430$ cm$^{-1}$ can be isolated. Yet, overlap with the induced absorption of $as$ distorts its lineshape and negative frequency-frequency correlations are not present for the entire signal. Overall, our key observations evidencing anti-correlated H-bonds are present for $D_2O$ in DMF and urea in DMSO. As such, the data in Fig. 5 support the notion that anti-correlated H-bonds are generic to H-bonding $XD_2/XH_2$ moieties.

## Conclusions

We find the H-bond distances of the two H-bond donating deuterium atoms of $D_2O$ and urea's $ND_2$ groups are anti-correlated: $D_2O$ and urea's $ND_2$ group preferentially donate one strong and one weak H-bond. This anti-correlation is encoded in the inhomogeneous linewidths of the decoupled and coupled O/N-D stretching modes and in frequency-frequency correlations between the asymmetric and

symmetric modes. Comparison to DFT calculations suggests that these anti-correlations stem from the H-bonding potential. Opposed to water in DMF, intermolecular coupling impedes spectroscopic detection of such anti-correlation in neat water[20], yet the H-bonding potentials for isolated water in DMF closely resemble the potential in neat water[45,46]. As such, similar H-bond conformations are likely present in liquid water. In fact, similar information on the H-bond correlations, herein obtained from the O-D cross-peaks, is in principle also contained in the O-D−O-H frequency correlations of a water molecule. Thus, using two different isotopic labels may make this methodology also applicable to pure water.

The anti-correlations are short-lived at ambient conditions and randomize in <500 fs–in line with predictions for water[16]. Our results highlight that H-bond asymmetry in water is not simply a statistical process. The static distribution of H-bonds and its dynamics are key to understanding the relation between H-bonded structure and the phase behavior of water. The dynamics are governed by low-frequency motions (H-bond stretching vibration and libration), and the formation of structurally different subphases of water must therefore be encoded in these low-frequency signatures. Our findings also have implications for understanding water as solvent, as the H-bonded structure and the spatial distribution of H-bonds around solutes and their lifetime may help understand water's ability to efficiently hydrate solutes.

## Methods

### Sample preparation

Deuteriumoxide ($D_2O$, 99.9 atom% D), urea (ACS reagent), and urea-d4 (98 atom% D) were purchased from Sigma-Aldrich and used without further purification. Dimethylsulfoxide (DMSO, 99.7+%, extra dry) and (DMF, 99.8%, extra dry) were purchased from Arcos Organics. The solvents DMF and DMSO were stored over 4 Å molecular sieve (Carl Roth), which was dried in a vacuum oven prior to use. $H_2O$ with a specific resistivity of 18.2 MΩ cm at 25 °C was obtained from a *Synergy* purification system (Merck). 10% $D_2O$/90% $H_2O$ and 100% $D_2O$ solutions of water in DMF were prepared volumetrically with a constant mole fraction of water of 5%. For the 10% $D_2O$/90% $H_2O$ mixture, HOD molecules comprise the major fraction of isotopically substituted species: a binomial distribution of isotopes results in 1% $D_2O$, 18% HOD, and 81% $H_2O$. Urea/urea-d4 mixtures (30% urea-d4 and 100% urea-d4) were prepared by weight in glass vials using an analytical balance. After isotopic equilibration the 30% urea-d4 corresponds to ~1% urea-d4, ~8% urea-d3, ~26% urea-d2, ~41% urea-d1, and ~24% urea. The total urea mole fraction was kept constant at 2.8%. To minimize water contamination for the urea/urea-d4 mixtures in DMSO, samples were prepared in an Ar-filled glovebox. To ensure isotopic equilibration, samples were prepared at least 24 h prior to experiments (see Supplementary Fig. 10). All samples were held between two $CaF_2$ windows separated by a Teflon spacer (urea-d4 100%: 100 μm; urea-d4 30%: 300 μm; 10% $D_2O$: 200 μm and 100% $D_2O$: 50 μm). To avoid uptake of moisture for urea in DMSO, the sample cells were assembled and filled in a glovebox.

### Infrared absorption spectroscopy

Infrared spectra were recorded in transmission using a Bruker Vertex 70 IR spectrometer, with a resolution of 4 cm$^{-1}$ at frequencies ranging from 400 cm$^{-1}$ to 4000 cm$^{-1}$. The spectrometer was purged with dried air during measurement.

### Two-dimensional infrared spectroscopy

2D-IR experiments were based on 800 nm pulses (7 W, 35 fs, 1 kHz) from a Ti: sapphire-based regenerative amplifier (Coherent *Astrella*). A fraction of these pulses (pulse energy 2.7 mJ) was used to pump an optical parametric amplifier (Coherent *Topas Prime*) to generate signal and idler pulses. IR pulses at 4 μm (30 μJ, 400 cm$^{-1}$ FWHM) were

generated via non-collinear difference frequency generation between the signal and idler beams in a GaSe crystal (Coherent, *Topas*). The 4 μm pulses are sent to a commercial 2D-IR spectrometer *2D-Quick IR* (PhaseTech Spectroscopy, Inc.). The reflection at a wedged ZnSe window is used as a probe beam. Excitation pulses are generated from the residual IR beam in a pulse shaper, in which the IR beam is diffracted from a grating (200 l/mm), collimated by a parabolic mirror, and guided through an acousto-optic modulator (AOM). The beam is diffracted from the AOM and focused by a second parabolic mirror on a second grating (200 l/mm). The waiting time ($T_w$) of the excitation pulses, relative to the probe pulse, is controlled using a translational stage, and the polarization of the excitation beam is set to 45°, relative to the probe beam using a half-wave plate. Pump and probe pulses are focused by a parabolic mirror and overlapped at the sample position. After re-collimation with a second parabolic mirror, the probe beam components perpendicular/parallel to the pump pulse are transmitted/reflected through/from a wire grid polarizer, spectrally dispersed (SP2156 imaging spectrograph, Princeton Instruments, 30 l/mm grating), and detected (128 × 128-pixel mercury cadmium telluride array detector), to obtain the signals perpendicular ($\Delta\alpha_\perp(\tilde{\nu}_{probe})$) and parallel ($\Delta\alpha_\parallel(\tilde{\nu}_{probe})$) to the excitation polarization, respectively, as a function of detection frequency, $\tilde{\nu}_{probe}$, in the frequency domain. The excitation frequency, $\tilde{\nu}_{pump}$, is resolved in the time domain by generating two pump pulses that are delayed by $t_1$ using the pulse shaper. The resulting pulses have a pulse length of ~90 fs. A rotating frame at 2400 cm$^{-1}$ was used to reduce the number of time steps[47] and the raw data (700 fs at increments of 35 fs) were apodized using a Hamming window and zero-padded to 128 data points before Fourier transformation to the frequency domain (excitation frequency resolution of ~4 cm$^{-1}$). For better comparability, signals were Fourier-filtered analogously along the probe axis. All 2D-IR spectra were corrected for the frequency-dependent pump pulse intensity. From these data sets, isotropic 2D-IR spectra, $\Delta\alpha_{iso}(\tilde{\nu}_{pump}, \tilde{\nu}_{probe})$, which are free of orientational dynamics, were constructed using the Eq. (1):

$$\Delta\alpha_{iso}(\tilde{\nu}_{pump}, \tilde{\nu}_{probe}) = \frac{\Delta\alpha_\parallel(\tilde{\nu}_{pump}, \tilde{\nu}_{probe}) + 2\Delta\alpha_\perp(\tilde{\nu}_{pump}, \tilde{\nu}_{probe})}{3} \quad (1)$$

Figures 2b–e and 5a, b in the main manuscript show $\Delta\alpha_{iso}(\tilde{\nu}_{pump}, \tilde{\nu}_{probe})$ data, Figs. 4a, b and 5d show $\Delta\alpha_\perp(\tilde{\nu}_{pump}, \tilde{\nu}_{probe})$.

### DFT calculations

All calculations were performed using Orca 4.1.1[48] (BLYP[49,50] -D3(BJ)[51,52] /def2-TZVPP[53,54] level of theory). Geometries of one water (urea) molecule, donating two (four) H-bonds to two DMF (four DMSO) molecules embedded in a polarizable continuum[55] (DMF or DMSO) were optimized, with two hydrogen bond distances H$_{water}$···O$_{DMF}$ (H$_{urea}$···O$_{DMSO}$) of water (one amine group) constrained to 1.5−2.4 Å at increments of 0.1 Å (water) and to 1.5−2.4 Å at increments of 0.1 Å (urea). Coupled OD stretching frequencies of water were obtained using $D_2O$ and uncoupled stretching frequencies using HOD. For urea, coupled N-D stretching vibrations were calculated for one of urea's amine groups of urea-d2t2, with the hydrogen-bonding distances to the $ND_2$ groups being varied. Uncoupled N-D frequencies, were obtained based on urea-h1d1t2. All frequencies discussed in the manuscript are based on harmonic normal mode frequencies and were interpolated using spline interpolation. As anharmonic corrections exhibit a linear correlation with harmonic frequencies[56], we scale the thus obtained harmonic frequencies by a constant scaling factor for each mode such that the harmonic frequencies at the energetic minimum match the experimental center frequencies.

## Data availability

All data required to evaluate the conclusions of the paper are present in the paper and/or Supplementary Information. All data are available

from the corresponding author upon request. Source data are provided with this paper. Coordinate files of the DFT optimized geometries are provided as Supplementary Data 1. Source data are provided with this paper.

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

## Acknowledgements
We thank Carlos Baiz, Giulia Giubertoni, Ellen Backus, and Yuki Nagata for fruitful discussions. This project was supported by the European Research Council (ERC) under the European Union's Horizon 2020 research and innovation program (Grant Agreement 714691). B.A.M. thanks the Alexander von Humboldt Foundation for funding via a postdoctoral scholarship. Support from the MaxWater Initiative of the Max Planck Society is gratefully acknowledged.

## Author contributions
L.G., A.A.E., and J.H. conceptualized the study. L.G., A.A.E., M.B., M.G., and J.H. developed the methodology and analysis. L.G., A.A.E., C.S.K., B.A.M., and J.H. performed the experiments and computations. J.H. supervised the project. L.G. and A.A.E. wrote the original draft and L.G., A.A.E., C.S.K., B.A.M., M.B., M.G., and J.H. revised and edited the manuscript.

## Funding

## Competing interests
The authors declare no competing interests.
