## [Transparent Peer Review file · Nature Communications]

Dynamic anti-correlations of water hydrogen-bonds

Corresponding Author: Dr Johannes Hunger

Version 0:

Reviewer comments:

Reviewer #1

(Remarks to the Author)

This is an excellent paper, and I highly recommend publication. The goal of the paper is to determine the correlation/anti-correlation of the strength of the two hydrogen bonds a given molecule can form. The author, very convincingly, show that they are anti-correlated, albeit on very short timescales. The conclusion might very well resolve some of the very controversial conclusions that have been drawn from x-ray absorption and scattering experiments (which reveal a static snapshot picture). 2D IR spectroscopy on water, nowadays, is pretty standard and not particularly difficult. However, extremely good signal-to-noise is needed to elucidate the very small effect of positive vs negative correlation of the cross peak between the symmetric and the asymmetric stretch vibration. My feeling tells me that it is fact quite hard to do that in a reliable fashion. In fact, Fig. 4b, which is truly remarkable, looks almost too good to me, but I have no reason to doubt it. The paper is well written, and easy to follow, I would hope also for the non-2D-IR expert. I have only a few minor comments/questions the authors might want to consider:

-First the killer question (which is not supposed to be a killer question): After understanding the approach the authors took, the obvious change would be: Rather measuring the correlation/anticorrelation between the symmetric/asymmetric stretch vibration of D₂O, why not that between the localized OH and OD stretch vibrations of HOD? The former are mixed states, and it occurs to me that this cancels out correlations to a significant extent. Any thoughts on that? I feel the question is so obvious, that the authors should say something about it.

-Anticorrelated cross peaks have first been described in early papers of Hochstrasser, which probably should be cited here. The work was on different molecular system, but the idea is very much the same.

- The sentence "As such, the spread of H-bonding environments results in a broader distribution of frequencies for I as compared to as and sym, which is the predominant cause for the differing widths in Fig. 2a." is a bit misleading, in my opinion. That is, D₂O sits in the same "H-bonding environment". The rest of the paper explains very much in detail what the difference is, but that sentence is a bit superficial.

- As already said above, Fig. 4b looks almost too good. I would suggest to move one spectrum equivalent to Fig. 4a, where the correlation is positive, from SI to the main paper. After all, the negative correlation and the fact that it is very short-lived is the major conclusion of the paper.

- I don't understand Fig. 4c. First, I don't understand how the experimental data are generated, and second, why the calculated contour lines are not symmetric with respect to d₁ and d₂. What breaks the symmetry? Some spontaneous symmetry breaking and the internal degrees of freedom of the DMF molecules respond to it?

Overall, I congratulate the authors for this beautiful work

Reviewer #2

(Remarks to the Author)

The two hydrogen atoms of a water molecule can form hydrogen bonds of different strengths with the oxygen atoms of neighboring water molecules. The resulting complex H-bond network is probably reflected in the numerous structural, dynamic and thermodynamic anomalies of water. Gunkel et al. take a close look at this asymmetry using two-dimensional

infrared spectroscopy (2D-IR) supported by density functional theory (DFT) calculation on H-bond complexes.

Compared to the symmetric and asymmetric OD stretching vibrations of D2O, the inhomogeneous line width of the OD band in HDO clearly indicates that water exhibits noticeably anti-correlated H-bonds and thus forms two hydrogen bonds of different strengths.

Gunkel et al. demonstrated that the anti-correlated H-bonds are modulated by thermal motions of water on the sub-picosecond time scale. Thereby, the anti-correlations are consequences of the typical H-bonding potential of XH₂ groups, also observed for the ND₂ group of urea. For that purpose, the authors diluted water in dimethylformamide (DMF) for isolating the water molecules from each other. This way they could observe varying inhomogeneous broadening of the O-H stretching frequencies depending on the H-bond strength. The 2D-IR spectroscopy allowed showing that the observed anti-correlation persists for only a few hundred femtoseconds.

So far, there was no experimental evidence for the asymmetry of the H-bonds. In this respect, the investigations by Gunkel et al. are new and contribute significantly to the understanding of the H-bond network and its temporal fluctuations. The experimental methods used are very well suited to solving the problem.

What about the DFT calculations? Are they carried out in harmonic approximation? Is this sufficient for an adequate description of the complex OH vibrational modes? Or, do the authors have to take anharmonic behavior into account?

Should we not naturally expect anti-correlated hydrogen bonds in liquid water? Are such experiments also possible in hexagonal ice, in which the two hydrogen atoms are equally strong bound in H-bonds and these anti-correlations are not to be expected?

Isn't there already evidence for anti-correlated H-bridges in water clusters from IR experiments in the gas phase or in liquid helium?

The authors write in their introduction about a relationship between anti-correlated H-bonds and anomalous behavior of water. Can't they at least give an obvious example? After all, the question is whether H-bonds of different strengths and anti-correlations that are only evident on the femtosecond time scale should really have an influence on anomalous properties at macroscopic level?

Overall, I suggest publication of the clearly and nicely written manuscript in Nature Communication. However, the posed questions should be properly addressed.

Reviewer #3

(Remarks to the Author)

In this manuscript, the authors investigate the water hydrogen-bonding using vibrational spectroscopy. The authors use a previously developed approach, consisting on studying the O-D stretching vibrations with IR and 2D-IR spectroscopy to probe the water hydrogen-bond. By focusing on both deuterated water (D₂O) and partially deuterated water (HOD) in dimethylformamide, the authors observe broader inhomogeneous linewidths for the OD band of HOD compared to the symmetric and asymmetric OD stretching modes of D₂O. In addition, the cross peaks observed in the spectra for D₂O provides evidence for the anti-correlations between the frequencies of the two vibrational modes of water, highlighting their modulation by the thermal motions of water on a sub-picosecond timescale. The authors use the difference in bandwidth and cross peak anti-correlation to infer information about the hydrogen bonds formed by water molecules and dimethylformamide. In addition, the authors use density functional theory calculations to support their experimental observations. Finally, the authors extend their analysis beyond water by investigating the H-bonding potential of XH₂ groups in other molecules such as urea. Their findings suggest a direct correlation between the structural arrangement of XH₂ groups and the observed anti-correlation in hydrogen bond distances. While the data is of good quality, the interpretation of the results is does not seem to be properly supported by vibrational theory. Therefore, the manuscript cannot be recommended for publication in its present form.

Major concerns:

- The authors compared the O-D of HOD, with the symmetric and asymmetric O-D stretch modes of D₂O to obtain the inhomogeneous bandwidth. This comparison is not correct because the O-D stretch of HOD is in the site representation and O-D stretches of D₂O is in the exciton state representation. In the latter case, the resulting frequencies are

$$\omega_{OD_as} = \omega_{OD} - ((d\omega_1 - d\omega_2)^2 + 4\beta^2)^{0.5} + d\omega_1 + d\omega_2$$

$$\omega_{OD_sym} = \omega_{OD} + ((d\omega_1 - d\omega_2)^2 + 4\beta^2)^{0.5} + d\omega_1 + d\omega_2$$

while in the former is only

$$\omega_{OD} + d\omega.$$

Clearly, these two frequencies are not comparable. Moreover, one could claim that the coupling constant, β , is negligible, so one can represent D₂O in the single site representation; but in this case, it is closed to ~50 cm⁻¹. It is therefore not possible to compare the inhomogeneous width of these two bands because they do not represent the same thing.

- Regarding the observed cross peak anti-correlation, the previous point is still valid. The cross correlation of the frequencies is not easily defined from the site representation because it includes both site frequency fluctuations and fluctuations of the coupling constant. Thus, it is hard to believe that the anti-correlation of the frequencies in the exciton representation denotes those of the sites.

- The authors also claim that water forms a strong and a weak hydrogen bond. It is not clear how the authors make such a connection. In the manuscript, the authors first claim that the anti-correlation in the CLS arises from the anti-correlated H-bond distances, which it is reasonable according to the data presented. However, the provided data do not give any

information about the thermodynamics of the H-bond in such a system.

- The authors omit very important citations in this manuscript. For example, the D2O as an isolated molecular system was previously study by the Fayer group (see doi: 10.1021/jp310086s). In this work, the CLS of the cross peak also oscillates. However, this work is not mentioned at all in the manuscript.

Minor comment:

- On page 9, the author describes the CLS dynamics of the cross peak with an exponential decay of 50fs. This appears to be a typo since this decay constant does not match figure 4b, where the decay occurs on a time scale of hundreds of femtoseconds.

Version 1:

Reviewer comments:

Reviewer #1

(Remarks to the Author)

I think the authors have adequately addressed my comments and recommend that the paper should now be published

Reviewer #2

(Remarks to the Author)

I have read the three reviews and the authors' response to them very carefully. In any case, I can say that the reviewers' concerns, critical questions and comments were answered comprehensively and meaningfully.

Where necessary, Gunkel et al. made changes and additions to the manuscript or the supplementary material. The literature relevant to this work was supplemented for the fundamental studies by Hochstrasse and Fayer.

Overall, Gunkel et al. have addressed the reviewers' criticism in an excellent manner, so that I can now unreservedly recommend the publication of the manuscript in Nature Communications.

Reviewer #3

(Remarks to the Author)

The authors have successfully addressed my comments and concerns, so I can now recommend the paper for publication.

Reviewer #1

This is an excellent paper, and I highly recommend publication. The goal of the paper is to determine the correlation/anti-correlation of the strength of the two hydrogen bonds a given molecule can form. The author, very convincingly, show that they are anti-correlated, albeit on very short timescales. The conclusion might very well resolve some of the very controversial conclusions that have been drawn from x-ray absorption and scattering experiments (which reveal a static snapshot picture). 2D IR spectroscopy on water, nowadays, is pretty standard and not particularly difficult. However, extremely good signal-to-noise is needed to elucidate the very small effect of positive vs negative correlation of the cross peak between the symmetric and the asymmetric stretch vibration. My feeling tells me that it is fact quite hard to do that in a reliable fashion. In fact, Fig. 4b, which is truly remarkable, looks almost too good to me, but I have no reason to doubt it. The paper is well written, and easy to follow, I would hope also for the non-2D-IR expert. I have only a few minor comments/questions the authors might want to consider:

We thank the reviewer for carefully reading our manuscript and for their very positive and encouraging comments.

First the killer question (which is not supposed to be a killer question): After understanding the approach the authors took, the obvious change would be: Rather measuring the correlation/anticorrelation between the symmetric/asymmetric stretch vibration of D₂O, why not that between the localized OH and OD stretch vibrations of HOD? The former are mixed states, and it occurs to me that this cancels out correlations to a significant extent. Any thoughts on that? I feel the question is so obvious, that the authors should say something about it.

The reviewer raises an excellent point. Indeed, frequency cross-correlations between OH- and OD-stretching modes of the same water molecule would provide the same information on H-bond correlations. In fact, experiments using two heavy isotopes could reduce contributions due to the heated ground state to the spectra and potentially provide these correlations also in liquid water. However, such experiments would require a broadband 2D-IR or two-color 2D IR experiment. Unfortunately, we do not currently have the experimental capabilities to perform such experiments, but we plan to implement this in the future. Also, experiments using two heavy isotopes are currently not possible. We have therefore added a comment on such experiments as an outlook to the revised manuscript, which reads:

"In fact, similar information on the H-bond correlations, herein obtained from the O-D cross-peaks, is in principle also contained in the O-D – O-H frequency correlations of a water molecule. Thus, using two different isotopic labels may make this methodology also applicable to pure water."

Anticorrelated cross peaks have first been described in early papers of Hochstrasser, which probably should be cited here. The work was on different molecular system, but the idea is very much the same.

We thank the reviewer for bringing the similarity of these earlier papers by Hochstrasser to our work to our attention. We fully agree that credit should be given to these seminal works. We have included these references (Refs. 33-36 of the revised manuscript), which reads:

"The correlation between $\tilde{\nu}_{sym}$ and $\tilde{\nu}_{as}$ can be directly interrogated with 2D-IR spectroscopy as coupling between sym and as gives rise to the cross-peak ($\tilde{\nu}_{pump} \approx 2540 \text{ cm}^{-1}$ / $\tilde{\nu}_{probe} \approx 2620 \text{ cm}^{-1}$) in Figure 2c and the line shape of the cross-peak provides information on the correlations of H-bonding distances, analogously to earlier studies by Hochstrasser and coworkers³³⁻³⁶ on different molecular systems."

The sentence "As such, the spread of H-bonding environments results in a broader distribution of frequencies for l as compared to as and sym, which is the predominant cause for the differing widths in Fig. 2a." is a bit misleading, in my opinion. That is, D₂O sits in the same "H-bonding environment". The rest of the paper explains very much in detail what the difference is, but that sentence is a bit superficial.

We thank the reviewer for bringing this point to our attention, and we fully agree that D₂O and HOD experience the same environment, yet the interaction of both local O-D modes in D₂O gives rise to the differing linewidths. We have revised this sentence to make this clearer. The sentence now reads:

"As such, although D₂O and HOD experience very similar H-bonding environments, coupling of the two O-D stretching modes in D₂O results in a narrower distribution of frequencies for as and sym as compared to l, which is the predominant cause for the differing widths in Figure 2a."

As already said above, Fig. 4b looks almost too good. I would suggest to move one spectrum equivalent to Fig. 4a, where the correlation is positive, from SI to the main paper. After all, the negative correlation and the fact that it is very short-lived is the major conclusion of the paper.

According to the reviewer's suggestion, we have included the spectrum at 250 fs waiting time in Figure 4 of the main text of the revised manuscript.

Due to the reviewer's comment and in light of the reference by the Fayer group mentioned by reviewer 3, we discuss the origin of the oscillatory behavior in the revised manuscript in more detail. For metal carbonyls and water in ionic liquids, for which inhomogeneous broadening is much less pronounced than for water in DMF, similar oscillations have been observed in the 2D-IR peak intensities and also in the CLS dynamics. These oscillations have been ascribed to coherence transfer within the decoherence

time t_1 of the time domain 2D-IR experiment. This scenario is, however, rendered unlikely for the present case because we do not find evidence for modulation of the signal intensities at the same frequency. We also have performed further experiments in which we only excited the symmetric stretching mode. In these experiments, coherence transfer from the asymmetric stretch to the symmetric stretch is suppressed, however, as shown in Supplementary Figure 5 of the revised manuscript, the oscillations of the CLS remain present. Additionally, also the CLS dynamics of HOD in DMF, which exclusively probe the dynamics of the local mode and which cannot contain pathways with coherence transfer, exhibit a similar oscillatory pattern. As such, our findings suggest that the oscillations observed here stem from the inherent molecular dynamics of water (i.e., the lower frequency H-bond stretching and libration dynamics). We have included this discussion in the main text of the revised manuscript, which reads:

“Remarkably, $CLS(T_w)$ appears to decay with marked oscillatory dynamics. It has previously been reported that such oscillatory dynamics may stem from coherence transfer within the time interval between the two excitation pulses in the time-domain 2D-IR experiment.^{38,39} This scenario is, however, rendered unlikely for the present system because (i) we find no evidence for oscillations in the 2D-IR signal intensities (Supplementary Figure 3), (ii) 2D-IR spectra with suppressed coherence transfer from as to sym exhibit the same oscillatory CLS dynamics (Supplementary Figure 5), and (iii) also the CLS dynamics of the diagonal peak for I exhibit oscillatory dynamics with the same oscillation period (Supplementary Figure 6). As such, these observations suggest that the oscillations of the frequency-frequency correlations in Figure 4c rather originate from a modulation of the CLS due to the inherent dynamics of water, which is also supported by molecular dynamics simulations.²⁷”

I don't understand Fig. 4c. First, I don't understand how the experimental data are generated

We apologize for not clearly explaining the approach how the distribution of H-bond conformations $P(d_1, d_2)$ were obtained. The experimental data correspond to the purely inhomogeneous distribution of frequencies, which we obtain from the analysis of the 2D-IR cuts: We take the purely inhomogeneous linewidth $\Gamma_{G,i}$ and the center frequency $\tilde{\nu}_{c,i}$ listed in the Supplementary Information to obtain the distribution of frequencies $P_i(\tilde{\nu})$ (Figure 4e of the revised manuscript). Based on this distribution we take $P(d_1, d_2)$ as a fit parameter and optimize $P(d_1, d_2)$ such that the distribution of frequencies $P_i(\tilde{\nu})$ obtained from the frequency maps $\tilde{\nu}(d_1, d_2)$ and their probability $P(d_1, d_2)$ matches the experimentally inferred $P_i(\tilde{\nu})$ for all three vibrational modes.

To better explain this approach, with which we use the information from all three vibrational modes to estimate the distribution of H-bond geometries, we have expanded the description in the main text of the revised manuscript, which we hope is now easier to rationalize:

“To elucidate the origin of the H-bond anti-correlation, we estimate the distribution of H-bond conformations, $P(d_1, d_2)$, (Figure 4d) by simultaneously using the information from all vibrational modes I, sym, as. Therefore, we take the purely inhomogeneous distribution of frequencies, $P_i(\tilde{\nu})$, using the center frequencies and the purely inhomogeneous linewidths ($\Gamma_{G,i}$) obtained from analysis of the diagonal and anti-diagonal lineshapes in Figure 2 (see Supplementary Note 2 and Supplementary Tables 2-3) together with the frequencies of all modes as a function of H-bond geometry $\tilde{\nu}_i(d_1, d_2)$ (frequency maps in Figure 3a). For numerical treatment, we discretize $P_i(\tilde{\nu})$ at intervals of 15 cm^{-1} (Figure 4e, symbols) and the population of H-bond geometries $P(d_1, d_2)$ at intervals of 0.01 \AA (Figure

4d). Starting from random distributions of H-bond conformations $P(d_1, d_2)$, we optimize $P(d_1, d_2)$ such that the distributions $P_i(\tilde{\nu})$ calculated from the conformation distribution and the maps in Figure 3a match the discretized, purely inhomogeneous distribution of frequencies, $P_i(\tilde{\nu})$ in Figure 4e (details on the numerical accuracy are given in Supplementary Note 8 and Supplementary Figure 7).”

and second, why the calculated contour lines are not symmetric with respect to d_1 and d_2 . What breaks the symmetry? Some spontaneous symmetry breaking and the internal degrees of freedom of the DMF molecules respond to it?”

The reviewer spotted an important point, which we did not explain in detail in our original submission. Indeed, the calculated minimum potential energy is located at $d_1 \neq d_2$. This asymmetry originates from simplifying water dissolved in DMF by only three molecules in the DFT calculations (i.e. neglecting interaction with the surrounding bath). To accurately predict H-bond frequencies with DFT, we use dispersion corrections for the DFT calculations. However, these dispersion corrections also include interactions between the two DMF molecules, and their interaction is not fully symmetric. This interaction between the two DMF molecules gives rise to the global minimum of calculations of $\text{H}_2\text{O}+2$ DMF being slightly displaced from $d_1 = d_2$.

To evidence dispersion corrections being the origin of the asymmetry, we performed DFT calculations without dispersion corrections, which indeed exhibit a fully symmetric H-bond potential. We show these calculations in Supplementary Figure S8 of the revised submission. We emphasize that while the exact location of the center of the $P(d_1, d_2)$ distribution can depend on such details of the calculations, the shape of the distribution (evidencing the anti-correlated H-bond distances) is remarkably insensitive to these details. We discuss these aspects and the origin of this minor asymmetry in the main text of the revised manuscript, which reads:

“The distribution of H-bond distances thus obtained are shown in Figure 4d. We note that the slight displacement of the maximum of the distribution in Figure 4d from the diagonal (symmetric H-bond distances) results from the simplified representation of water in DMF by only one water molecule and two DMF molecules in the DFT calculations used to obtain the frequency maps in Figure 3a (dispersive interactions between two DMF molecules lead to symmetry breaking, see Supplementary Figure 8).”

Reviewer #2

The two hydrogen atoms of a water molecule can form hydrogen bonds of different strengths with the oxygen atoms of neighboring water molecules. The resulting complex H-bond network is probably reflected in the numerous structural, dynamic and thermodynamic anomalies of water. Gunkel et al. take a close look at this asymmetry using two-dimensional infrared spectroscopy (2D-IR) supported by density functional theory (DFT) calculation on H-bond complexes.

Compared to the symmetric and asymmetric OD stretching vibrations of D₂O, the inhomogeneous line width of the OD band in HDO clearly indicates that water exhibits noticeably anti-correlated H-bonds and thus forms two hydrogen bonds of different strengths. Gunkel et al. demonstrated that the anti-correlated H-bonds are modulated by thermal motions of water on the sub-picosecond time scale. Thereby, the anti-correlations are consequences of the typical H-bonding potential of XH₂ groups, also observed for the ND₂ group of urea. For that purpose, the authors diluted water in dimethylformamide (DMF) for isolating the water molecules from each other. This way they could observe varying inhomogeneous broadening of the O-H stretching frequencies depending on the H-bond strength. The 2D-IR spectroscopy allowed showing that the observed anti-correlation persists for only a few hundred femtoseconds.

So far, there was no experimental evidence for the asymmetry of the H-bonds. In this respect, the investigations by Gunkel et al. are new and contribute significantly to the understanding of the H-bond network and its temporal fluctuations. The experimental methods used are very well suited to solving the problem.

We thank the reviewer for the very positive and encouraging comments.

What about the DFT calculations? Are they carried out in harmonic approximation? Is this sufficient for an adequate description of the complex OH vibrational modes? Or, do the authors have to take anharmonic behavior into account?

We thank the reviewer for this remark. All frequencies obtained from the DFT calculations are indeed harmonic frequencies. Anharmonic corrections would slightly shift these frequencies. Yet, as for instance demonstrated by Buczek et al. (Ref. 56 of the revised manuscript), calculated harmonic and anharmonic OH stretching frequencies exhibit a perfect linear correlation. As such, anharmonic corrections do not affect our qualitative conclusions from the DFT calculations shown in Figure 3 of the main manuscript. Further, this linear correlation implies that for our quantitative comparison of the DFT-calculated frequencies with the experimental results, a constant scaling factor, which we use for this comparison, suffices to capture the distribution of H-bond conformations, as shown in Figures 4d and e.

We have clarified the use of harmonic frequencies and the impact of anharmonic corrections in the methods section of the revised manuscript, which reads:

“All frequencies discussed in the manuscript are based on harmonic normal mode frequencies. As anharmonic corrections exhibit a linear correlation with harmonic frequencies,⁵⁶ we scale the thus obtained harmonic frequencies by a constant scaling factor for each mode such that the harmonic frequencies at the energetic minimum match the experimental center frequencies.”

Should we not naturally expect anti-correlated hydrogen bonds in liquid water?

We understand that the reviewer refers to different measures for correlations. This is indeed an excellent point, and we realized that this was not discussed in detail in our original submission. If the H-bond energies of H-bond 1 and H-bond 2 were fully independent, the minimum energy pathways in the potential energy landscape $E(d_1, d_2)$ would simply parallel the d_1 and d_2 axis (at the equilibrium H-bond distances), and the two dissociation paths would be orthogonal. Due to the anharmonicity of the H-bond potential, the distribution of H-bond distances for an individual H-bond is inherently asymmetric. As such, the distribution $P(d_1, d_2)$ for two uncorrelated H-bonds would be inherently **asymmetric** (L-shaped distribution). Yet, the H-bonds would **not be anti-correlated** (the distribution of H-bond distances d_1 would still be independent of d_2). Such asymmetric distribution would indeed give negative Pearson correlation coefficients for the distribution of H-bonds, however, the Pearson correlation coefficient is only meaningful for a symmetric (Gaussian) distribution of two variables – as opposed to the inherent asymmetric distribution of H-bond distances. As such, one might trivially expect negative correlation coefficients for $P(d_1, d_2)$ distributions, which are, however, a mere result of the inadequacy of these correlation coefficients.

The distribution shown in Figure 4 and the potential energy shown in Supplementary 8, however, show that the energetically most favorable H-bond distance of one H-bond depends on the distance of the other H-bond. That is, H-bond energies are (anti-)correlated. We emphasize this fact explicitly in the revised manuscript, which reads:

“Assuming a Boltzmann distribution, anti-correlated H-bonds are also predicted solely based on the DFT-calculated energy of $D_2O + 2 DMF$: The shape of the Boltzmann distribution obtained from DFT (contour lines in Figure 4d) and the estimated distribution $P(d_1, d_2)$ (pixel plot in Figure 4d) agree well. The differing widths of both distributions are likely due to neglecting repulsive interactions with other molecules and overestimation of the homogeneous linewidth. Irrespective of these different widths, both distributions show that the energetically most favorable H-bond distance for one H-bond markedly depends on the H-bond distance of the other H-bond – the H-bonds are anti-correlated.”

Are such experiments also possible in hexagonal ice, in which the two hydrogen atoms are equally strong bound in H-bonds and these anti-correlations are not to be expected?

We agree that such experiments for different states of water would be highly desirable. Our insights into the H-bond conformations obtained here rely on isolating the effect of coupling between the two OD stretching modes within a single water molecule. In hexagonal ice (and also in neat water) intermolecular coupling gives rise to additional line broadening, which scrambles the effect of intramolecular coupling. As already mentioned in our response to reviewer 1, such experiments would be possible by using two heavy isotopes of hydrogen, yet we currently cannot perform such experiments. We therefore added this notion to the conclusion of the revised manuscript:

“In fact, similar information on the H-bond correlations, herein obtained from the O-D cross-peaks, is in principle also contained in the O-D – O-H frequency correlations of a water molecule. Thus, using two different isotopic labels may make this methodology also applicable to pure water.”

Isn't there already evidence for anti-correlated H-bridges in water clusters from IR experiments in the gas phase or in liquid helium?”

We agree that experiments on water clusters in noble gas droplets have provided valuable insights into the structure of water. Indeed, the inferred structures of such clusters (see, for instance, the nice overview by Saykally, doi: 10.1073/pnas.191266498) imply that water molecules tend to form H-bonds of different strengths. These conditions (a small water cluster embedded in a noble gas droplet) are arguably quite different from water in solution, as the interface between the cluster and the matrix makes the environment inherently asymmetric. For heterogeneous clusters, this notion seems also to hold (see e.g. doi: 10.1063/1.4979558). However, in our case, water is in a nominally symmetric environment, with two H-bond acceptors equally accessible to form H-bonds to water. We are unaware of any experiments in liquid helium where this nominal symmetric environment would be similarly present. Also, the presence of such anti-correlations at ambient temperatures, which is the focus of the present study, would still remain elusive. As such, despite the detailed structural insight obtained from the infrared spectra of such clusters, we feel that the environment of water and the thermally accessible dynamics differ in these experiments from our study. Therefore, the results and the conclusions are not directly comparable.

The authors write in their introduction about a relationship between anti-correlated H-bonds and anomalous behavior of water. Can't they at least give an obvious example? After all, the question is whether H-bonds of different strengths and anti-correlations that are only evident on the femtosecond time scale should really have an influence on anomalous properties at macroscopic level?

The asymmetry of water's H-bonded network has been related to the anomalous properties and our results demonstrate that this asymmetry, in part, results from anti-correlated H-bond distances. As such, our results help rationalize the origins of this asymmetry. Yet, our studied system, isolated water molecules in DMF, is too simple to make firm predictions on the implications of the anti-correlations for liquid water. Therefore, we hesitate to speculate about the consequences of our findings for the anomalous properties of water. Here, experiments studying liquid water would certainly be required.

Nevertheless, we have elaborated on the implications that have been reported in the literature in the introduction of the revised manuscript:

“Such motifs have been suggested to have profound implications for the phase behavior of water¹⁰ and may explain some of the anomalous properties of water at reduced temperatures, such as a density maximum at 277 K or a nonlinear heat capacity temperature relationship.^{11,12”}

And describe the methodology that would be required to study liquid water in the conclusions of the revised manuscript:

“In fact, similar information on the H-bond correlations, herein obtained from the O-D cross-peaks, is in principle also contained in the O-D – O-H frequency correlations of a water molecule. Thus, using two different isotopic labels may make this methodology also applicable to pure water”

Overall, I suggest publication of the clearly and nicely written manuscript in Nature Communication. However, the posed questions should be properly addressed.

We thank the reviewer for the positive assessment and hope that we have addressed his/her comments adequately.

Reviewer #3

In this manuscript, the authors investigate the water hydrogen-bonding using vibrational spectroscopy. The authors use a previously developed approach, consisting on studying the O-D stretching vibrations with IR and 2D-IR spectroscopy to probe the water hydrogen-bond. By focusing on both deuterated water (D2O) and partially deuterated water (HOD) in dimethylformamide, the authors observe broader inhomogeneous linewidths for the OD band of HOD compared to the symmetric and asymmetric OD stretching modes of D2O. In addition, the cross peaks observed in the spectra for D2O provides evidence for the anti-correlations between the frequencies of the two vibrational modes of water, highlighting their modulation by the thermal motions of water on a sub-picosecond timescale. The authors use the difference in bandwidth and cross peak anti-correlation to infer information about the hydrogen bonds formed by water molecules and dimethylformamide. In addition, the authors use density functional theory calculations to support their experimental observations. Finally, the authors extend their analysis beyond water by investigating the H-bonding potential of XH2 groups in other molecules such as urea. Their findings suggest a direct correlation between the structural arrangement of XH2 groups and the observed anti-correlation in hydrogen bond distances. While the data is of good quality, the interpretation of the results is does not seem to be properly supported by vibrational theory. Therefore, the manuscript cannot be recommended for publication in its present form."

"Major concerns:

- The authors compared the O-D of HOD, with the symmetric and asymmetric O-D stretch modes of D2O to obtain the inhomogeneous bandwidth. This comparison is not correct because the O-D stretch of HOD is in the site representation and O-D stretches of D2O is in the exciton state representation. In the latter case, the resulting frequencies are

$$\omega_{OD_{as}} = \omega_{OD} - ((d\omega_1 - d\omega_2)^2 + 4\beta^2)^{0.5} + d\omega_1 + d\omega_2$$

$$\omega_{OD_{sym}} = \omega_{OD} + ((d\omega_1 - d\omega_2)^2 + 4\beta^2)^{0.5} + d\omega_1 + d\omega_2$$

while in the former is only

$$\omega_{OD} + d\omega.$$

Clearly, these two frequencies are not comparable. Moreover, one could claim that the coupling constant, β , is negligible, so one can represent D2O in the single site representation; but in this case, it is closed to ~ 50 cm⁻¹. It is therefore not possible to compare the inhomogeneous width of these two bands because they do not represent the same thing.

We fully agree with the reviewer that the linewidths of the local stretching mode and the coupled stretching modes have different origins and are, therefore, not directly comparable. The coupling described by the reviewer is indeed why we find the linewidths to differ, and it is indeed not

straightforward to discriminate between line broadening due to fluctuations of the hydrogen bond and due to coupling.

We emphasize that we, therefore, use the frequencies obtained from DFT calculations to analyze and interpret our data. These calculations account for changes in normal mode frequencies due to variations in the H-bond strength (via variation of the H-bond distance). Also, these calculations intrinsically contain coupling between local modes, and, thus, account for changes of normal mode frequencies produced by variation of the coupled local mode frequencies. As such, both contributions described above ($\Delta\omega$ and β) are accounted for in these calculations. The calculations in Figure 3 demonstrate that coupling effects largely vary, depending on the H-bond distances of both sites. In fact, these calculations show that due to the subtle interplay between local site fluctuations and coupling effects, the differing linewidths contain information on the H-bond distributions: correlated and anti-correlated H-bond distances give rise to a different line broadening of the symmetric/asymmetric modes of D₂O and local O-D mode of HOD. In our manuscript, we use these differences in linewidths to estimate the distribution of H-bond distances (shown in Figure 4d of the revised manuscript). From this analysis, we find the H-bond distances to be anti-correlated.

To better illustrate the information contained in these linewidths, we calculated the distribution of H-bonds analogously to Figure 4d, yet assuming as a hypothetical example that the linewidths of all three vibrational modes are the same (30 cm⁻¹). As can be seen from Figure R1, for this fictitious example, our approach would predict correlated H-bond distances. This result illustrates that the very different line-broadening contributions indeed contain information on the H-bond configurations.

Figure R1: (left) Fitted H-bond conformation distribution $P(d_1, d_2)$ of D₂O in DMF obtained from minimizing deviations between the experimental, discretized inhomogeneous distribution of frequencies $P_i(\tilde{\nu})$ (symbols in right panel) and the calculated distribution $P(d_1, d_2)$ together with the frequency maps in Figure 3a of the main manuscript. $P_i(\tilde{\nu})$ in the right panel assume a hypothetical linewidth of 30 cm⁻¹ for *l*, *as*, and *sym*.

To avoid potential confusion, we state in the introduction of the revised manuscript that, due to their different nature, the linewidths of the coupled and uncoupled modes may differ:

“The instantaneous frequencies of sym and as depend on the instantaneous frequencies of l1 and l2, thus on the H-bond distances of the two O-H groups, and on the coupling strength (Figure 1c). These different factors governing the instantaneous frequencies of sym, as, l1 and l2 make also the linewidths of coupled and uncoupled modes to differ. In turn, the correlation of H-bonds can be probed via the lineshapes of sym and as.”

To better emphasize that we isolate the coupling contribution using DFT, we explain the calculations of the harmonic frequencies in more detail in the revised manuscript:

“To understand the origin of these different inhomogeneous widths, we calculated normal mode frequencies under the harmonic approximation for different H-bond geometries using DFT. We specifically address the effect of coupling on the harmonic normal mode frequencies by calculating the frequencies for l , sym , and as using H/D isotope exchange”

We further emphasize in the revised manuscript that we use the information from the differing linewidths to extract the H-bond distance distributions:

“To elucidate the origin of the H-bond anti-correlation, we estimate the distribution of H-bond conformations, $P(d_1, d_2)$, (Figure 4d) by simultaneously using the information from all vibrational modes l , sym , as .”

Regarding the observed cross peak anti-correlation, the previous point is still valid. The cross correlation of the frequencies is not easily defined from the site representation because it includes both site frequency fluctuations and fluctuations of the coupling constant. Thus, it is hard to believe that the anti-correlation of the frequencies in the exciton representation denotes those of the sites.

We agree that the frequency cross-correlation indeed depends on the complex interplay between the coupling of both OD oscillators and coupling to the bath. As such, frequency correlations of the coupled modes cannot easily be related to the local modes. However, also here, the DFT calculations, which account for both, local mode fluctuations and fluctuations of coupling, come to help. The calculations shown in Figure 3b,d show that for correlated and uncorrelated H-bond distances, the symmetric and antisymmetric stretching vibrations are correlated, while for anti-correlated H-bond distances, the coupled modes are anti-correlated (Figure 3c). We have added this notion to the revised manuscript:

“Although it is challenging to discriminate between ii) and iii) solely based on the frequencies of the three modes as a function of H-bond lengths, both limiting cases can be discerned by considering the relation between the DFT-calculated frequencies $\tilde{\nu}_{sym}$ and $\tilde{\nu}_{as}$: For ii) the slopes of $\tilde{\nu}_{sym}(d_1)$ and $\tilde{\nu}_{as}(d_1)$ have opposite sign (Figure 3c): Upon increasing d_1 , $\tilde{\nu}_{sym}$ increases (decreases) and $\tilde{\nu}_{as}$ decreases (increases) for $d_1 \leq 1.9 \text{ \AA}$ ($d_1 \geq 1.9 \text{ \AA}$). Conversely, for iii) $\tilde{\nu}_{sym}(d_1)$ and $\tilde{\nu}_{as}(d_1)$ have positive or zero slopes throughout (Figure 3d). Hence, for anti-correlated distances d_1 and d_2 (ii), the frequencies $\tilde{\nu}_{sym}$ and $\tilde{\nu}_{as}$ are anti-correlated, while for uncorrelated distances (iii) $\tilde{\nu}_{sym}$ and $\tilde{\nu}_{as}$ are correlated.”

The authors also claim that water forms a strong and a weak hydrogen bond. It is not clear how the authors make such a connection. In the manuscript, the authors first claim that the anti-correlation in the CLS arises from the anti-correlated H-bond distances, which it is reasonable according to the data presented. However, the provided data do not give any information about the thermodynamics of the H-bond in such a system.

We fully agree with the reviewer’s comment. We used the terms H-bond length and H-bond strength interchangeably in our original submission, which is not correct. Most of our analysis is based on H-

bond lengths, while information on the H-bond energetics is only implicitly shown by the calculated DFT energies. Accordingly, we have revised this terminology throughout the manuscript and replaced H-bond strength by H-bond length/distance wherever appropriate.

The authors omit very important citations in this manuscript. For example, the D2O as an isolated molecular system was previously studied by the Fayer group (see doi: 10.1021/jp310086s). In this work, the CLS of the cross peak also oscillates. However, this work is not mentioned at all in the manuscript.

We thank the reviewer for pointing out this important reference, and we apologize for overseeing this highly relevant paper. We now refer to the paper in the main text of the revised manuscript (Ref. 38 of the revised manuscript). In this very nice work by Fayer and co-workers, the interaction of water molecules with their environment (ionic liquid) is rather weak. Therefore, inhomogeneous line broadening is less pronounced, and the study thus provides in-depth insights into the coupling of the two OD groups. Conversely, for water isolated in DMF, as studied in our manuscript, coupling to the bath (i.e. H-bonding interactions) and coupling of the OD- modes is comparable. As such, our experiments enable detailed insights into H-bonding.

As suggested by Fayer et al. (Ref. 28 of the revised manuscript and similarly by Kubarych et al. for systems with very homogeneous linewidths in Ref. 39 of the revised manuscript), coherence transfer may be consistent with some aspects of our observations (the oscillatory dynamics of the center line slopes). This scenario is, however, rendered unlikely for the present case because – unlike in these references – we do not find evidence for modulation of the signal intensities at the same frequency. Further, we have performed experiments exciting only the symmetric stretching mode. In these experiments, coherence transfer from the asymmetric stretch to the symmetric stretch is suppressed. However, as shown in Supplementary Figure 5 of the revised manuscript, the oscillations of the CLS are hardly altered. Additionally, also the CLS dynamics of HOD, which probe only the local mode, and therefore, coherence transfer between the two coupled modes cannot occur, exhibit a similar oscillatory pattern. Together, these findings suggest that the oscillations stem from the inherent molecular dynamics of water (i.e. the lower frequency H-bond stretching mode and librational dynamics). We have included this discussion in the main text of the revised manuscript, which reads:

“It has previously been reported that such oscillatory dynamics may stem from coherence transfer within the time interval between the two excitation pulses in the time-domain 2D-IR experiment.^{38,39} This scenario is, however, rendered unlikely for the present system because (i) we find no evidence for oscillations in the 2D-IR signal intensities (Supplementary Figure 3), (ii) 2D-IR spectra with suppressed coherence transfer from as to sym⁴⁰ exhibit the same oscillatory CLS dynamics (Supplementary Figure 5), and (iii) also the CLS dynamics of the diagonal peak for I exhibit oscillatory dynamics with the same oscillation period (Supplementary Figure 6). As such, these observations suggest that the oscillations of the frequency-frequency correlations in Figure 4c rather originate from a modulation of the CLS due to the inherent dynamics of water, which is also supported by molecular dynamics simulations.²⁷ “

Minor comment:

On page 9, the author describes the CLS dynamics of the cross peak with an exponential decay of 50fs. This appears to be a typo since this decay constant does not match figure 4b, where the decay occurs on a time scale of hundreds of femtoseconds.”

We apologize for potential confusion. We fit the data with a sum of a damped oscillation (representative of the H-bond stretching vibration) and an exponential decay (representative of librations). As pointed out by the reviewer, the damping time is markedly longer ($\tau_{\text{damp}} = 470$ fs) than the exponential decay time $\tau_{\text{ex}} = 50$ fs (see also Supplementary Table 7). To avoid any ambiguity, we have added the damping time to the caption of Figure 4 of the revised manuscript:

“Center-line slope dynamics, $CLS(T_w)$ of the off-diagonal peak. Symbols show experimental data and error bars show the uncertainty of the slope obtained from linear regression of the center points. The solid line shows a fit using a sum of a damped oscillation (~310 fs oscillation period, 470 fs damping time) and an exponential decay (50 fs decay time).”